# Protein interface redesign facilitates the transformation of nanocage building blocks to 1D and 2D nanomaterials

Xiaorong Zhang[1], Yu Liu[1], Bowen Zheng[1], Jiachen Zang[1], Chenyan Lv[1], Tuo Zhang [1✉], Hongfei Wang[2] & Guanghua Zhao [1✉]

Although various artificial protein nanoarchitectures have been constructed, controlling the transformation between different protein assemblies has largely been unexplored. Here, we describe an approach to realize the self-assembly transformation of dimeric building blocks by adjusting their geometric arrangement. *Thermotoga maritima* ferritin (TmFtn) naturally occurs as a dimer; twelve of these dimers interact with each other in a head-to-side manner to generate 24-meric hollow protein nanocage in the presence of $Ca^{2+}$ or PEG. By tuning two contiguous dimeric proteins to interact in a fully or partially side-by-side fashion through protein interface redesign, we can render the self-assembly transformation of such dimeric building blocks from the protein nanocage to filament, nanorod and nanoribbon in response to multiple external stimuli. We show similar dimeric protein building blocks can generate three kinds of protein materials in a manner that highly resembles natural pentamer building blocks from viral capsids that form different protein assemblies.

[1] College of Food Science & Nutritional Engineering, China Agricultural University, Beijing Key Laboratory of Functional Food from Plant Resources, Beijing 100083, China. [2] Key Laboratory of Chemical Biology and Molecular Engineering of Education Ministry, Key Laboratory of Energy Conversion and Storage Materials of Shanxi Province, Institute of Molecular Science, Shanxi University, Taiyuan, China. ✉email: zhangtuo@cau.edu.cn; gzhao@cau.edu.cn

Shape transformation phenomena are ubiquitous in nature. Many living organisms by shape transformation perform shape-to-function activities in response to the external environment[1]. For instance, *Amoeba proteus* undergoes multi-directional shape transformation to form pseudopod for navigation and rapid path alteration. These phenomena have triggered tremendous interest in mimicking the structure–property relationship of living systems. At a molecular level, shape-shifting related to DNA[2], RNA[3], peptides[4], and small molecules[5] has been reported in recent years owing to their relatively simple and controllable structure. In viral capsids, a single protein fold can be evolved to form multiple oligomeric states with different symmetries[6], but to construct smart protein architectures artificially whose structure and shape transformation could be modulated by external stimuli remains challenging.

Proteins, as Nature's most versatile building blocks, are mainly responsible for the complexity of living organisms[7]. During evolution, proteins have acquired self-assembly properties to construct a variety of large, complex, and symmetric architectures such as one-dimensional (1D) actin filaments[8], two-dimensional (2D) bacterial surface layers (S-layers)[9], and three-dimensional (3D) light-harvesting protein complexes of phycobilisomes[10], thereby endowing their hosts with plenty of functions. It is well known that protein–protein interactions (PPIs) at protein interfaces are the chief contributors to construct the diversified protein nanostructures[11–13]. Following Nature's inspiration to assemble protein building blocks into exquisite nanostructures, various self-assembly strategies, such as symmetry-directed design[14–17], metal coordination[7,18–20], host–guest interactions[21,22], and the use of bifunctional ligands[23,24], have been applied to construct 1D, 2D, and 3D hierarchical protein nanostructures. Among these various protein nanostructures, natural protein nanocages represent a class of versatile nanomaterials that fulfill a wide range of functions, such as $CO_2$ fixation by carboxysomes[25], iron metabolism by ferritins[26], DNA protection by Dps[27], and nucleic acid storage and transport by viral capsids[28]. By taking advantage of their well-defined architectures, isolated interiors, and high biocompatibility, scientists have subverted the above natural functions of the protein nanocages and explored them as nano-containers for encapsulation and delivery of bioactive cargo molecules[29], as bio-templates for preparation of various nanomaterials[30], and as reaction centers for multienzyme catalysis[31]. So far, different strategies such as de novo design[32–34], fusion protein[14,35,36], directed evolution[37–39], and key interface redesign[40,41] have been built to create a variety of artificial hollow protein nanocages that rival the size, property, and functionality of their natural analogs. Despite these advances, rendering PPIs controllable to facilitate the transformation of the building blocks from protein nanocages into 1D or high-order nanomaterials in the laboratory has yet to be explored.

Herein we introduce a protein interface redesign approach that could be used for the self-assembly transformation of dimeric building blocks from hollow protein nanocage to filament, nanorod, and nanoribbon. The basic building block of this approach is a naturally occurring dimeric protein—*Thermotoga maritima* ferritin (TmFtn), which tends to assemble into 24-meric hollow protein nanocage induced by calcium ions (Fig. 1a). The crystal structure analyses revealed that the assembly of two adjacent dimeric protein molecules in a head-to-side manner is responsible for the formation of such a shell-like structure (Fig. 1a). In contrast, upon tuning such head-to-side state to a fully side-by-side manner by protein interface redesign, similar dimeric protein molecules self-assemble into filaments or nanorods in the presence of calcium ions or polyethylene glycol (PEG), respectively (Fig. 1b). Differently, when we adjusted the two adjacent dimeric proteins to interact in a partially side-by-side manner by protein interface redesign, the self-assembly of these similar dimeric protein molecules are able to transform from the inherent hollow protein nanocage into nanoribbon in the presence of PEG (Fig. 1c). The dimensions of these filaments, nanorods, and nanoribbons collectively span between nanometer and micrometer scales (50 nm to 4.0 μm). In-depth characterization by X-ray crystallography, transmission electron microscopy (TEM), and atomic force microscopy (AFM) confirmed that this protein interface redesign approach can regulate the transformation between hollow protein nanocage and 1D and 2D nanomaterials. This approach opens up an avenue for constructing 1D or 2D nanoarchitectures with the building blocks of hollow protein nanocage as starting materials.

## Results

**Natural dimeric TmFtn assembly into hollow protein nanocages.** The four helix bundle structure, which is widely distributed in Nature, has been utilized as building blocks to construct a number of proteins and enzymes to perform a considerably wide range of functions, such as iron storage by ferritin[26], DNA protection by Dps protein[27], copper storage by Csp1[42], electron transfer by cytochrome $cb_{562}$[43], ribonucleotide reduced into deoxynucleotide by R2 subunit of ribonucleotide reductase[44], methane oxidized into methanol by methane monooxygenase[45], and so on. Four helix bundles are also attractive for synthetic chemists because its interfaces are dominated by side chain and side chain interactions, which can be more tunable than β-strands. This study focuses on TmFtn, a naturally dimeric protein consisting of two antiparallel four helix bundles, as shown in Supplementary Fig. 1. In contrast, most known ferritins are usually composed of 24 identical or similar subunits that assemble into a shell-like structure. Interestingly, the dimeric TmFtn can convert into 24-meric protein nanocage in the presence of divalent metal ions such as Mg(II) and Ca(II)[46,47]. Consistent with these recent findings, our results show that, in solution, purified TmFtn molecules (Supplementary Fig. 2) exist as dimers and can self-assemble into 24-meric hollow protein nanocage in the presence of $Ca^{2+}$, and addition of EDTA causes the formed protein nanocage disassembly back into its dimeric form (Supplementary Fig. 3a), indicative of a reversible process of protein assembly (Supplementary Fig. 3b).

The crystal structure of 24-meric Tmftn determined at a resolution of 2.2 Å (Supplementary Tables 1 and 2) shows that one calcium ion located nearby $C_3$–$C_4$ interface is coordinated with two acidic residues (Glu51 and Glu132) and three water molecules (Supplementary Fig. 3c, d). Careful analyses of the crystal structure revealed that a group of acidic amino acid residues are lined along with the $C_3$–$C_4$ interface, producing electrostatic repulsion along with this interface in the absence of calcium ions. This might be an important reason why TmFtn molecules naturally exist as protein dimers in solution in the absence of metal ions. In contrast, calcium ions near the $C_3$–$C_4$ interface can essentially eliminate such electrostatic repulsion through their interaction with the acidic residues, strengthening the stability of $C_3$–$C_4$ interface, finally producing the 24-meric protein nanocage. Besides, each protein nanocage is composed of 12 protein dimers, and any of the 2 adjacent protein dimers interact with each other in a head-to-side manner (Supplementary Fig. 3b, d). Based on these findings, it is reasonable to believe that calcium ions facilitate the interaction of these protein dimers in a head-to-side manner, resulting in the formation of hollow protein nanocage.

**Engineering dimeric TmFtn for assembly into filaments and nanorods.** The above head-to-side interaction manner is

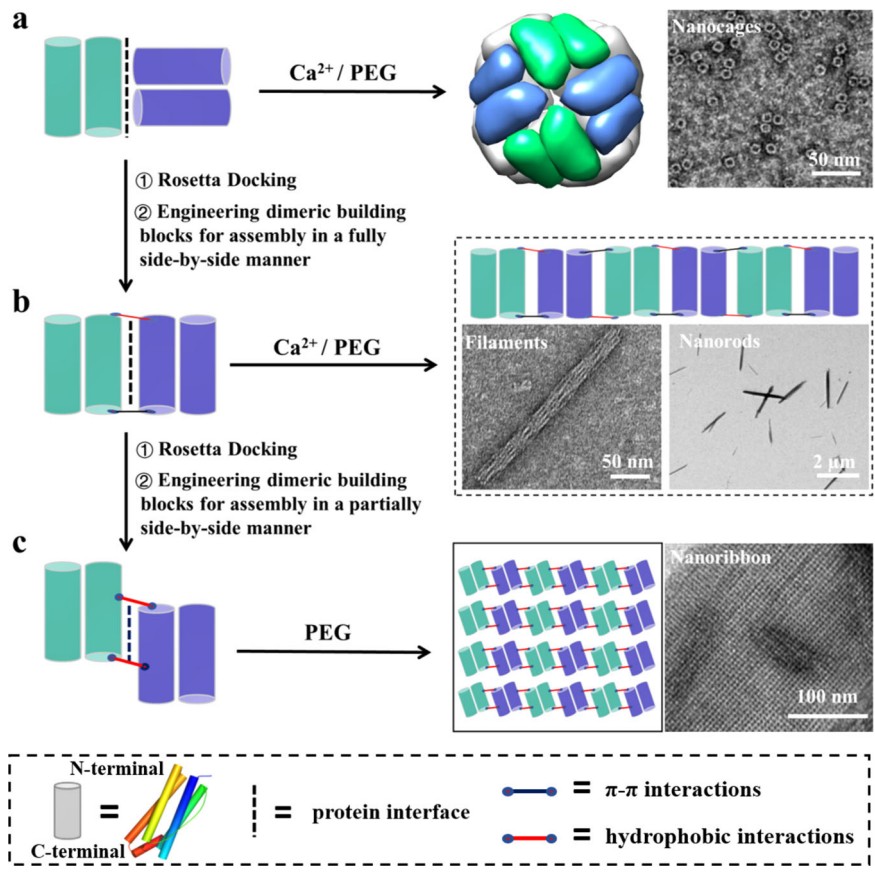

**Fig. 1 Schematic representation of the protein interface redesign approach used to control the self-assembly transformation of dimeric building blocks between different types of nanomaterials. a** The naturally occurring dimeric *Thermotoga maritima* ferritin (TmFtn) molecules interact with each other in a head-to-side manner to form 24-meric protein nanocages in the presence of calcium ions or PEG. **b**, **c** Upon Rosetta docking and designing new PPIs between adjacent dimeric building blocks in a fully or partially side-by-side manner, the assembly of similar dimeric building blocks can be transformed from the above formed hollow protein nanocages into nanofilaments or nanorods or nanoribbons in the presence of calcium ions or PEG.

ubiquitous in nearly all known ferritins from animal, plant, to bacteria. Based on this interesting phenomenon, we envisioned that if this head-to-side interaction manner between two adjacent dimeric TmFtn was adjusted to a fully side-by-side manner by protein interface redesign, the 24-meric TmFtn nanocages would transform into 1D protein nanomaterials. Therefore, we describe a three-step computational assisted method for realizing this idea: (1) two adjacent dimers were manually placed in a fully side-by-side manner as shown in Supplementary Fig. 4a, (2) RosettaDock algorithm[48] was carried out to optimize such fully side-by-side interactions into a more complementary protein–protein interface, and (3) building interactions at the designed interface to drive self-assembly. After completing the first two steps, predicted surface residues include sites 114 and 147 and sites 114' and 147' that lie across the designed interface from each other as appropriate locations for installing the interactions (Supplementary Fig. 4b, c). We made a mutant named FLAL where Asn147 was replaced by aromatic phenylalanine (Phe) residue to create $\pi-\pi$ interactions, while Glu112, Glu113, and Lys114 were replaced by three hydrophobic residues Leu, Ala, and Leu to decrease the electrostatic repulsion and increase the hydrophobic interactions between two adjacent dimers at the same time (Supplementary Fig. 5). Subsequently, this mutant was purified to homogeneity as characterized by sodium dodecyl sulfate (SDS) and native polyacrylamide gel electrophoresis (PAGE; Supplementary Fig. 6). However, similar to wild-type (wt) TmFtn, the mutant FLAL also occurs as a dimer rather than liner arrangement in solution under different experimental conditions (pH value from 6.0 to 10.0; the

concentration of NaCl from 0 to 500 mM; Supplementary Fig. 7). This might be because the designed interaction forces are not strong enough to drive protein dimers assembly in a side-by-side manner in solution. This situation is, at least partly, derived from the fact that a group of acidic amino acid residues from C-helix and D-helix occur between the designed subunits interface (Supplementary Fig. 8), reducing the designed attraction, thereby impeding adjacent FLAL molecule assembly in the designed side-by-side manner.

Inspired by the fact that $Ca^{2+}$ has the ability to induce natural TmFtn dimer assembly into 24-meric protein nanocage through its interaction with acidic residues (such interaction not only eliminates electrostatic repulsion from acidic residues but also reinforces the attraction between dimeric building blocks; Supplementary Fig. 3c), we deemed that calcium ions might also have the ability to facilitate FLAL molecules to assemble side-by-side because many acidic amino acids are lined along with the designed interface. To confirm this idea, we examined the FLAL self-assembly behavior in solution concerning $Ca^{2+}$ and FLAL concentrations. Upon screening the concentration of $Ca^{2+}$ and FLAL (Supplementary Table 3), we found that the optimal condition for the formation of filaments is [FLAL] = 48.0 μM and $[Ca^{2+}]$ = 80.0 mM, which is close to the $Ca^{2+}$ concentration (50 mM) used for the above nanocage assembly[47]. TEM under the above conditions showed that FLAL molecules self-assemble into two kinds of protein arrays: cuboid-like superlattices and filaments, as shown in Supplementary Fig. 9a. Since FLAL concentration used here is significantly lower than that needs

during conventional crystallization procedures (at least 144.0 μM or 6.0 mg/mL), the formation of these two kinds of protein arrays corresponds to a protein self-assembly process in solution. Enlargement of the small cuboid-like assembly revealed that 12 dimeric FLAL molecules first assemble into 24-meric protein nanocages, which further assemble into well-organized 3D superlattices (Supplementary Fig. 9b–d) as we previously reported[16,17]. This result is not surprising for two reasons: (a) the above genetic modification for native dimeric TmFtn is only involved in amino acid residues nearby the $C_3$ and $C_4$ interfaces of the metal-mediated 24-meric protein nanocage rather than its $C_3$–$C_4$ interfaces, thus hardly affecting its inherent assembly property; (b) substitution of Asn147 with Phe in native TmFtn can also have the possibility to make two adjacent protein nanocages join together along the fourfold channels through π–π interactions as reported recently[16], resulting in the formation of such 3D protein superlattices. In contrast, enlargement of the linear-shaped species revealed the generation of filaments by FLAL mutant molecules (Supplementary Fig. 9e). Differently, wt TmFtn dimers can only assemble into discrete protein nanocages under identical experimental conditions (Supplementary Fig. 10). These findings demonstrate that protein interface redesign in conjunction with appropriate solution conditions can realize the self-assembly transformation of dimeric building blocks from protein nanocage into 1D nanomaterials.

To visualize the morphology of the formed filaments in detail, they were observed by TEM in different visual fields. With excess $Ca^{2+}$ addition, filaments were formed rapidly within 30 min (Fig. 2a). As time goes on, the filaments become longer and more numerous (Fig. 2a–d), suggesting that the filament formation is thought to proceed through growth events. The length of these filaments grew up to tens of nanometers with the most extended filament up to 1.3 μm (Supplementary Fig. 11), and finally the formed filaments were tangled together (Fig. 2d). In addition, the filaments have a tendency to stack together, and thus individual filament is hardly observed. Usually, two or four filaments are stacked together, and these filament assemblies exhibit widths on the order of 7 or 17 nm, respectively, with the pitch of two filaments approximately 3 nm (Fig. 2e, f). The width of the individual filament is about 2 nm, which is in good agreement with the thickness of FLAL dimer (Supplementary Fig. 1).

To obtain detailed structural information on the FLAL filament, we tried to crystallize this protein and eventually obtained qualified single crystals suitable for X-ray diffraction. The crystal structure was solved at a resolution of 2.1 Å (Supplementary Tables 1 and 2). The packing pattern of FLAL molecules in the crystal is pronouncedly different from wt TmFtn. The side view of the crystal structure revealed that FLAL molecules arrange in a repeating, side-by-side fashion to form filaments (Fig. 3a). In contrast, native TmFtn exhibits a different packing pattern in its crystal where 12 TmFtn dimers assemble into a 24-mer protein cage (Supplementary Fig. 3). Such a difference in protein assembly between TmFtn and mutant FLAL agrees with our design. The crystal structure shows that the width of the filament is about 5 nm from the side view and about 2 nm from the top view, these findings being in accordance with the above TEM observation showing that the width of filaments is about 2 nm (Fig. 2).

Further crystal analyses reveal that two adjacent FLAL molecules have opposite orientations, forming another interface along their C and D helixes. In the structure with resolutions that permit detailed analysis of side-chain configurations, Phe147 and Leu114 side chains at the designed interface adopt suitable conformations to generate π–π stacking interactions and hydrophobic interactions, respectively (Fig. 3b, c), again approving our design at an atomic level. Besides the designed noncovalent interactions, Leu112 and Val119 also produce hydrophobic interactions (Fig. 3d), which is unexpected. Additionally, one calcium ion occurs nearby the designed protein interface, which binds to Glu108, Asp127, Asn105, and one water molecule (Fig. 3e). We believe that the cooperation of these noncovalent interactions and metal coordination along the inter-building-block interfaces promotes the formation of the filaments. It should be noted that only the bounded calcium ion with obvious densities can be identified in the crystal structure, but other interactions with low occupancy were not counted. In the crystal structure, the formed filaments further arrange in the vertical direction to create 2D protein assemblies (Supplementary Fig. 12a). Besides, these filaments are parallel displaced from the side view (Supplementary Fig. 12b), and two adjacent filaments are connected by weak interactions in the crystal structure. For example, Lys10 and Glu164 from two contiguous filaments are in close proximity, leading to the generation of electrostatic attraction (Supplementary Fig. 12c). However, TEM analyses revealed that the M1 mutant where Lys10 and Glu164 were mutated into Gly exhibits a similar assembly behavior to the

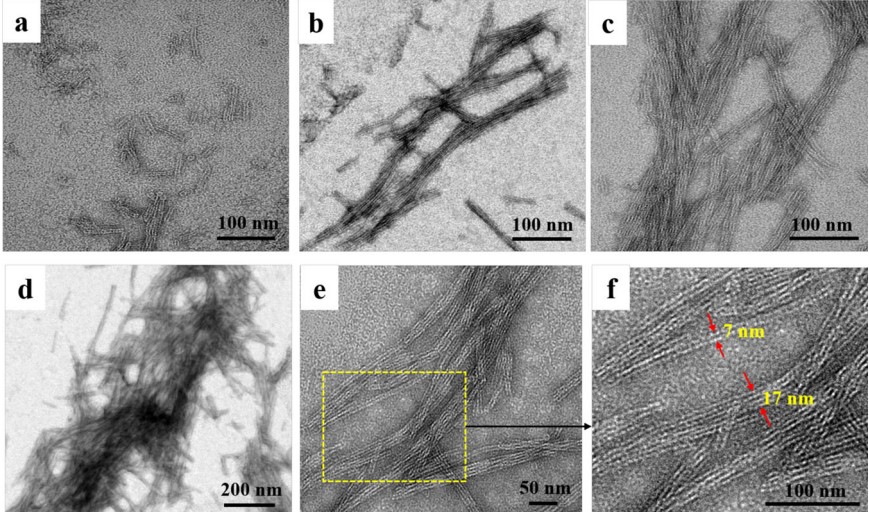

**Fig. 2 Characterization of the filaments constructed by 48.0 μM FLAL and 80 mM $Ca^{2+}$. a–d** TEM images of the filaments at different time points (0.5, 1, 12, and 24 h). **e** High-magnification transmission electron microscopic (HRTEM) images of the filaments. **f** Real map of the inverted FFT of **e**.

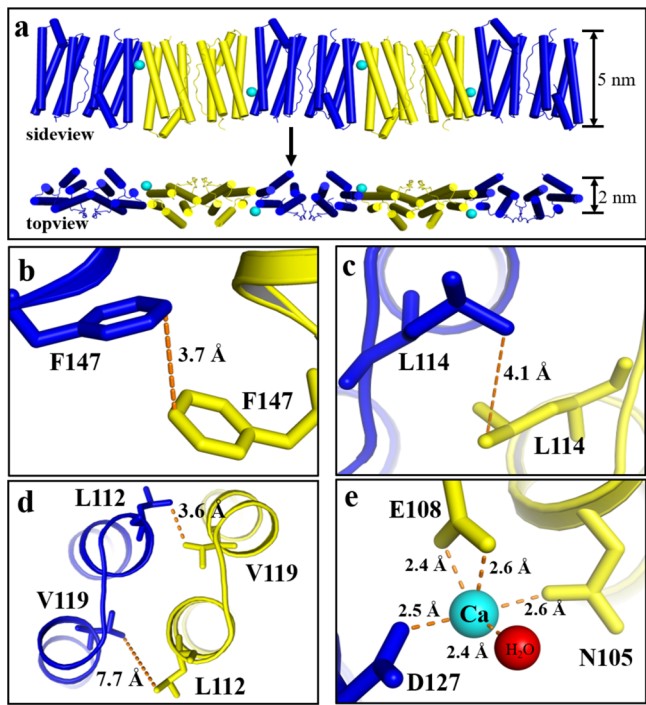

**Fig. 3 Structural basis of the 1D filaments in atomic detail. a** Side and top views of the FLAL filament in the crystal structure where the Ca$^{2+}$ is shown as cyan sphere. **b–e** Close-up views of the interfacial interactions between two adjacent dimers in the 1D filament, including $\pi$–$\pi$ stacking interaction (**b**), hydrophobic interactions (**c, d**), and metal coordination (**e**).

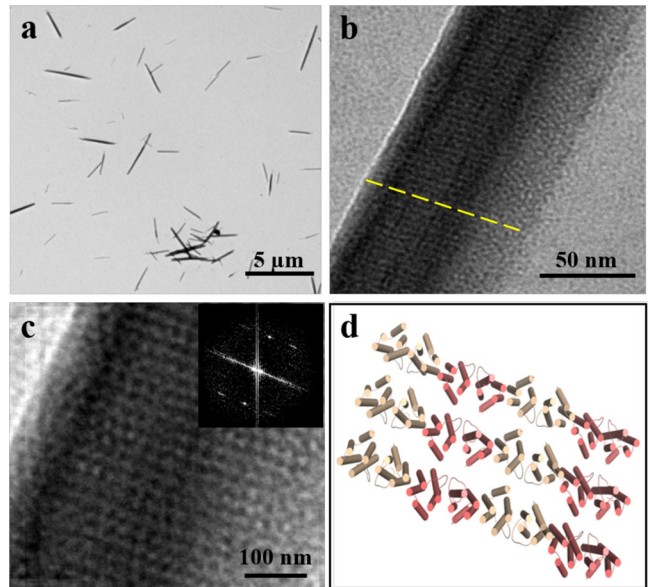

**Fig. 4 TEM analyses of the nanorods constructed by 12.0 μM FLAL and 30% PEG1500. a** Low-magnification TEM view of the formed nanorods. **b** High-magnification view of **a**. The yellow dotted line indicated the assembled direction of FLAL molecules. **c** Real map of the inverted FFT from **b**. Inset: Fast Fourier transform image. **d** Structural diagram based on the reconstruction of **c**.

FLAL molecule (Supplementary Fig. 13), suggesting that other kinds of non-covalent interactions might be also responsible for the formation of the observed 1D filament. The detailed stacking mechanism between adjacent filaments needs further investigation.

One overarching goal of synthetic biomimetic nanomaterials is to construct hierarchical assemblies that are able to respond to various external stimuli so that the fabricated assemblies can be controlled. Therefore, besides calcium ions, we also investigated the effect of PEG on the assembly behavior of FLAL molecules because PEG is a hydrophilic nonionic polymer widely used in many biochemical and pharmaceutical applications due to its mild action on the biological activity of cell components[49,50]. After trying different molecular weights (PEG1000, PEG1500, and PEG3000) and concentrations (10, 20, and 30%) of PEG, we finally obtained nanorods formed in the presence of 30% PEG1500. As shown in detail in Fig. 4a, nanorods of several micrometers in length (up to 4 μm) can be visualized through TEM. Magnification of the assembled components showed that FLAL molecules are well organized and arranged in a linear manner as shown by the yellow dotted line (Fig. 4b). Fast Fourier transform (FFT) based on the TEM image presents a clear view of the assembled arrangement (Fig. 4c). The corresponding assembly pattern of the FLAL molecules observed under TEM is shown in Fig. 4d. Further analysis by AFM with an intelligent mode revealed that the height of the nanorod is 220 ± 5 nm (Supplementary Fig. 14). In contrast, wt TmFtn dimers only assemble into discrete protein nanocages under identical experimental conditions (Supplementary Fig. 15). These results again demonstrate that the designed driving force enables dimeric FLAL molecules to self-assemble into nanorods.

To determine the structural basis of such nanorods, we set out to grow single crystals large enough for X-ray diffraction analyses in the presence of PEG1500. We solved the crystal structure at 2.6 Å resolution (Supplementary Tables 1 and 2). As expected, the designed interface is responsible for driving self-assembly, but no electron densities of the PEG were observed (Supplementary Fig. 16), most likely due to the high flexibility of PEG and its weak ability to bind to protein[50]. The crystal structure proved remarkably similar to the design model: protein molecules connect through designed $\pi$ stacking and hydrophobic interactions in a side-by-side repeating pattern with adjacent FLAL molecules antiparallel with each other (Supplementary Fig. 16a, e). Thus, the crystal structure is in good agreement with the above results observed by TEM (Fig. 4b, c), confirming the side-by-side interaction manner of dimeric FLAL molecules.

Further analyses of the crystal structure showed that the assembly process of FLAL molecules could be summarized as follows: first, FLAL molecules pack along the $x$-axis in such a way mentioned above to form 1D arrays (Supplementary Fig. 16a), and such assembly pattern was also observed by TEM shown in Fig. 4b; second, just like the stacking of filaments of FLAL molecules in the presence of Ca$^{2+}$, these 1D arrays stack along the $y$-axis through electrostatic interaction to form 2D arrays (Supplementary Fig. 16b), which is likewise visualized by TEM (Fig. 4b, c); finally, the formed 2D arrays further arrange in the vertical direction to create 3D protein assemblies (Supplementary Fig. 16c, d), and the weak interaction involved in this step mainly comes from their E helix (Supplementary Fig. 16f). Taken together, all these findings demonstrate that our protein interface redesign approach yields assembly of dimeric FLAL protein building blocks into the filaments or nanorods induced by Ca$^{2+}$ or PEG rather than the inherent Ca$^{2+}$-mediated 24-meric protein shell-like assembly.

**Engineering dimeric TmFtn for assembly into nanoribbons.** The above results demonstrated that tuning the action mode of two adjacent protein dimers from the head-to-side manner to the fully side-by-side manner facilitates the self-assembly transformation of the dimeric building blocks from hollow protein nanocage into nanofilaments and nanorods. Bioinspired by the

structure of α-keratin protofilaments formed from two staggered rows of head-to-tail coiled coils[51], we wonder what if we rearrange two adjacent protein dimers in a staggered pattern, namely, a partially side-by-side pattern. To answer this question, we use the method mentioned above for protein docking but do not need to preset the dimers to search for their possible arrangement. To this end, we simulated the orientations of two TmFtn dimers through global docking using RosettaDock[48]. One thousand independent docking trajectories were carried out, and by combining energy simulation with a visual inspection, a model with dimers in a partially side-by-side manner was selected to assist interface redesign (Supplementary Fig. 17a). We found that several hydrophobic residues, including Leu107, Val119, Val126, and Val130, are distributed along with the designed interface (Supplementary Fig. 17b). To effectively utilize these hydrophobic residues, we plan to design hydrophobic interactions as the driving forces to construct such partially side-by-side protein assembly. Based on the fact that FLAL mutant contains three more hydrophobic amino acid residues located on its outer surface than wt TmFtn, therefore, to build stronger hydrophobic interactions at this designed interface (Supplementary Fig. 17c), we chose FLAL mutant instead of wt TmFtn as starting materials for further engineering.

To enable two dimers to interact with each other in a partially side-by-side fashion, we made another mutant named FLAL-L where Arg137 in FLAL was replaced with Leu (Supplementary Fig. 5) to increase the area of the hydrophobic patch as shown in Supplementary Fig. 17d. To gain insight into the assembly behavior of FLAL-L mutant, we purified it to homogeneity as suggested by SDS-PAGE and native-PAGE (Supplementary Fig. 18). We next investigated the assembly of FLAL-L molecules in the presence of PEG in solution. Upon screening the concentration and molecular weight of PEG, we found that FLAL-L molecules can likewise assemble into filaments in the presence of 15 or 20% PEG1500, as shown in Supplementary Table 4. The filaments with different sizes ranging from 100 nm to several micrometers can be visualized (Fig. 5a, b). Enlargement of the assembly revealed that the FLAL-L molecules are well organized in a linear fashion (Fig. 5b). To obtain more insights into the structural information, the formed filaments were further characterized by AFM with an intelligent mode. As shown in Fig. 5c, d, the height of the array is $6 \pm 0.5$ nm, which is comparable with the height of FLAL-L molecules (~5 nm) in the crystal (Supplementary Fig. 1). To gain deep insight into the above process, we investigated the formation of the filaments as a function of time. Short filaments were rapidly formed at 5 min. With an increase in time, the length and number of the formed filaments increased gradually (Fig. 6a–e). After 15 days, except for the filament, nanoribbon-like nanomaterials appeared (Fig. 6f). A high-magnification TEM view (Fig. 6g) seized the periodic parallel arrangement of protein arrays, and the real map from invert FFT (Fig. 6h) gave an excellent view on the structure of the nanoribbon, which reveals that two kinds of lines interlaced to form such structure (Fig. 6i). Further examination of the nanoribbons using AFM revealed that the height of the nanoribbon is nearly 10 nm (Supplementary Fig. 19).

To shed light on structural information on the above filaments and nanoribbons, FLAL-L was also crystallized by vapor diffusion, and eventually, qualified single crystals suitable for X-ray diffraction were obtained (Supplementary Fig. 20a). We solved the crystal structure at the resolution of 2.3 Å (Supplementary Tables 1 and 2). The crystal structure analyses revealed that FLAL-L molecules are perfectly aligned with each other in a partially side-by-side manner to form filaments (Fig. 7a), the top view of which is in good agreement with the filament observed under TEM (Fig. 5a, b). As expected, intermolecular associations

between two adjacent FLAL-L molecules are mediated entirely through hydrophobic interactions, as the contact region of adjacent FLAL-L molecules is rich in hydrophobic amino acid residues (Fig. 7b). The engineered Leu114 and Leu137 side chains cooperated with the original V119, L107, V126, and V130 side chains provide the bulk of the hydrophobic core, whereas the engineered Leu112, Ala113, and Phe147 are not involved in such interactions. The crystal structure provides direct evidence to confirm the partially side-by-side manner of FLAL-L molecules. The above formed filaments coalesce in parallel along the y axes through hydrogen bonds (Supplementary Fig. 20b, c), leading to the generation of a protein layer (Fig. 7c), corresponding to the ribbons observed by TEM (Fig. 6). The arrangement of the protein layer in the crystal perfectly matches the pattern seen in the nanoribbons observed by TEM (Fig. 6f–i). Based on the observation that the thickness of each protein layer is about 2 nm as suggested by the crystal structure (Fig. 7c) and the height of the nanoribbons detected by AFM (Supplementary Fig. 19) is around ~10 nm, it is reasonable to believe that the observed nanoribbons are of five layers. In crystals, the formed protein layers further arrange along the z axes to form 3D protein frameworks (Fig. 7d) through π–π interactions and electrostatic interactions (Supplementary Fig. 20d–f). All these results demonstrate that the hydrophobic interactions between two contiguous dimeric protein-building blocks in a partially side-by-side fashion carried out by protein interface redesign facilitate the transformation of the dimeric building blocks from 24-meric nanocages into nanoribbons through filaments.

## Discussion

Particle polymorphism is a popular phenomenon in virus capsids, which possess inherent switches that allow efficient disassembly and reassembly in response to solution conditions. For example, an icosahedral plant virus, cowpea chlorotic mottle virus, can convert into a tubular nanostructure, which needs double-stranded DNA to act as a template and take advantage of the nonspecific binding of capsid proteins and DNA[52,53]. Simian virus 40 (SV40) represents another interesting example of shape transformation between protein nanostructures with different dimensions. The SV40 capsid is mainly composed of 72 pentamers of VP1. Electron microscopic observations revealed that, at pH 5.0, long and tubular structures are formed, whereas at high salt concentrations, small (20 nm) particles predominate[54]. However, how to transform the assembly of the building blocks from natural protein cages into 1D or 2D architecture by a simple, effective method in the laboratory has yet to be explored. The challenge lies in the fact that protein nanocages occurs as discrete molecules in solution while filament/nanorod/ribbon created in this study closely resemble 1D and 2D polymers and that the intermolecular interface and assembly geometry for the construction of hollow protein nanocage and filament/nanorod/ribbon nanomaterials are entirely different. We demonstrate that spatially directed assembly of protein-building blocks might be a solution to solve such problems, which will reduce the computational workload and lower the barriers to protein assembly design. To confirm this idea, we have built here a protein interface redesign approach that is able to manipulate the directed assembly of protein-building blocks, resulting in the self-assembly transformation of dimeric building blocks from hollow protein nanocage to 1D and 2D protein arrays with nanoscale and microscale long-range order.

In pursuit of our goal to transform the dimeric building blocks from hollow protein nanocages into 1D and 2D nanostructure, external stimuli were introduced to exert control over PPIs for dimeric FLAL or FLAL-L assembly. Researchers have

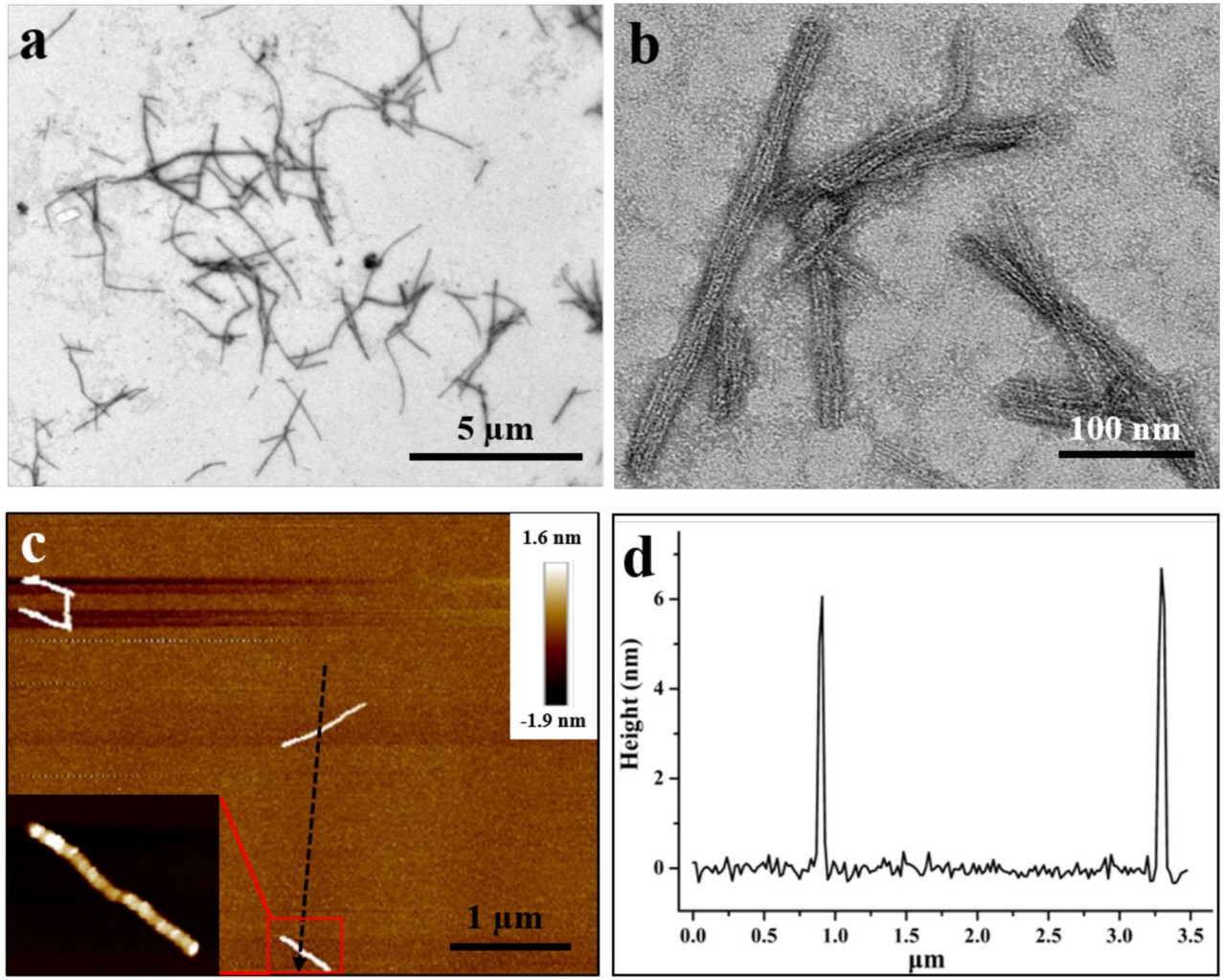

**Fig. 5 Characterization of the 1D filaments constructed by 12.0 μM FLAL-L and 15% PEG1500. a** Low-magnification TEM view of FLAL-L filaments. **b** Enlargement of FLAL-L filaments. **c** Atomic force microscopic (AFM) micrographs of the 1D filaments. **d** Height distributions measured along the black arrow in **c**.

demonstrated that the self-assembly of a monomeric four helix bundle protein can be directed in all three dimensions through the inherent chemically tunable and modular nature of metal coordination[7]. In this study, the transformation of dimeric building blocks into different protein nanomaterials including hollow protein nanocage and filament/nanorod/nanoribbon mainly depends on the redesigned PPIs. These PPIs are largely mediated by the designed weak, noncovalent interactions, although external stimulation is required to trigger the assembly of the designed protein dimers.

It is noteworthy that there is only one amino acid difference in amino acid sequence between FLAL and FLAL-L molecules, but their assembly behavior is markedly different from each other. FLAL molecules self-assemble into nanorods, whereas FLAL-L has the ability to assemble into the nanoribbons. The significant difference in self-assembly behavior between these two dimeric protein molecules reflects the importance of amino acid residue Arg137 in FLAL, suggesting that Arg137 could act as a switch to control the conversion of nanorods to nanoribbons. In all cases, the crystal structures reveal that the backbones in the side-by-side interaction manner were designed with high accuracy compared to the computationally designed model (Supplementary Fig. 21). The occurrence of 1D and 2D protein arrays with the dimeric protein as building blocks recalls among natural proteins the case

of actin filaments and S-layers in terms of their dimensions and structural uniformity. Thus, our reported construction of 1D and 2D protein arrays from the dimeric protein as building blocks provides a model to study the assembly mechanism of natural protein architectures.

Controlling self-assembly is critical to the advancement of nanotechnology. Nowadays, scientists are able to accurately control the protein self-assembly behaviors to construct various supramolecular structures through the rational design of PPIs. Though a variety of intricate protein nanostructures such as hollow protein cages, filaments/tubules, nanosheets, and 3D crystalline frameworks have been created[18–24,36–41], rendering directed assembly of protein-building blocks into the custom-tailored nanoarchitectures remains challenging. Our reported protein engineering approach could tune the inherent head-to-side interaction manner of two adjacent dimeric protein-building blocks to the fully or partially side-by-side manner by redesigning protein interfaces, yielding directed assembly of the building blocks, thereby facilitating the self-assembly transformation of the dimeric building blocks from hollow protein nanocage into 1D or 2D nanomaterials. PPIs at the designed protein interfaces are mainly contributed from the cooperation of the hydrophobic interactions, aromatic π–π interactions, and external stimulation, which effectively stabilize the designed protein interface. The

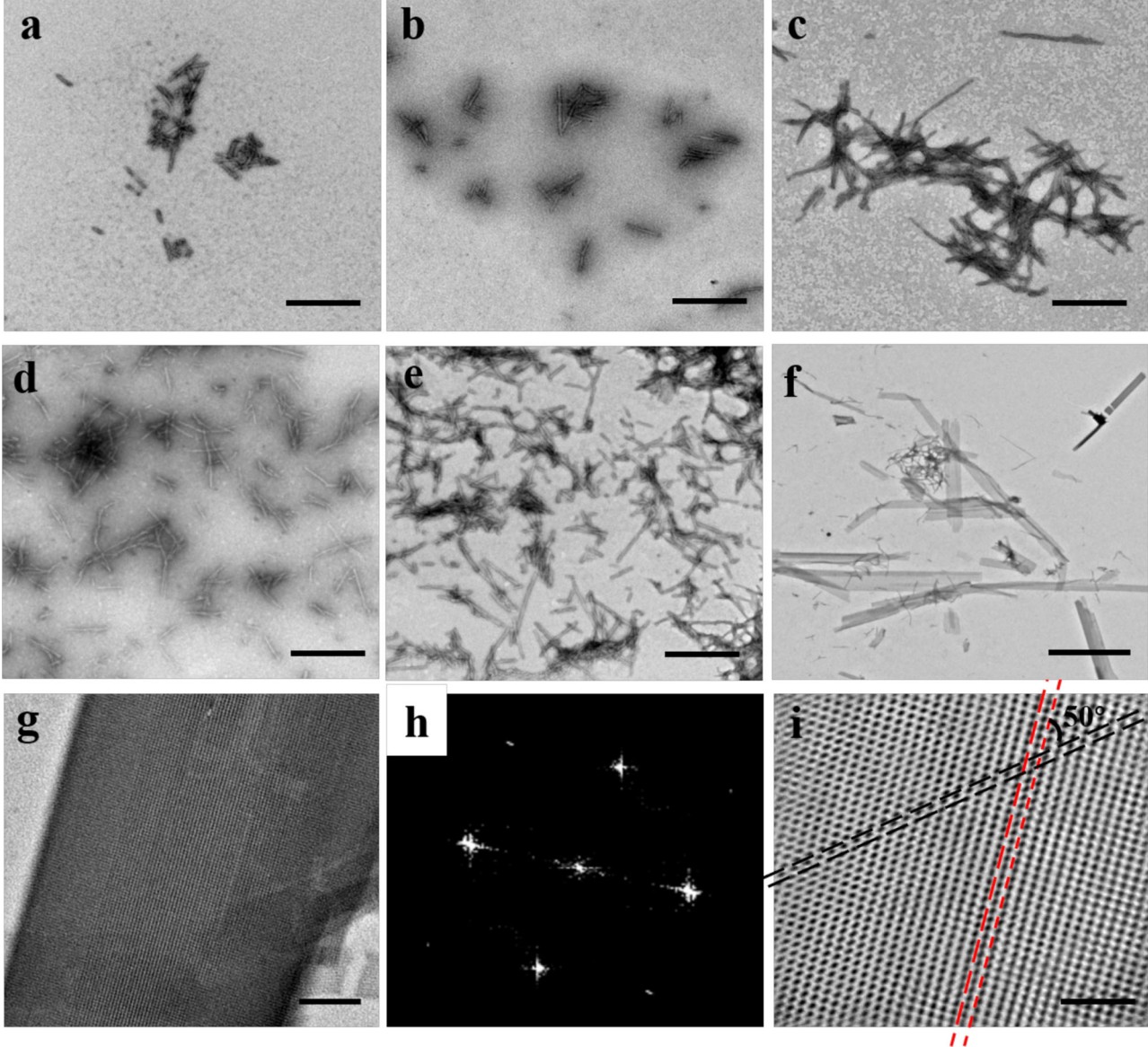

**Fig. 6 Kinetics of the formation of 2D nanoribbon. a–f** TEM views of the formation of FLAL-L assemblies as a function of time (5 min, 30 min, 1 h, 12 h, 24 h, and 15 days, respectively) in the presence of 15% PEG1500. **g** High-magnification view of the 2D nanoribbon formed at 15 days. **h** Fast Fourier transform (FFT) of **g**. **i** Real map from invert FFT of **h**. The black and red dotted lines indicate the arrangement of the 2D FLAL-L assembly. **a–e** Scale bars represent 500 nm, while scale bars represent 2 μm in **f**, and 100 nm in **g**, **i**, respectively.

above protein engineering approach that focuses on adjustment for the geometric arrangement of protein-building blocks is conceptually and operationally simple. It is interesting that the head-to-side interaction between two contiguous dimeric protein-building blocks occurs not only in all known ferritins but also in other cage-like proteins such as Dps, where six dimeric protein-building blocks also assemble into a shell-like structure[27]. Therefore, our engineering approach should, in principle, be applicable to some other protein architectures. This would produce a variety of protein nanomaterials with different geometries. Furthermore, the filaments obtained in this study could provide a multiple enzyme port scaffold to convene variant enzymes align into a supramolecular enzyme system with unexplored property[55].

## Methods

**Protein preparation**. The gene encoding TmFtn was synthesized by Synbio Technologies, which has been inserted into the plasmid pET-3a. Mutagenesis of the

TmFtn cDNA was performed with the fast site-directed mutagenesis kit (TIANGEN Biotech Co., Ltd.). Polymerase chain reaction amplification was carried out using the pET-3a plasmid with the TmFtn gene as a template, and primers for mutagenesis are listed in Supplementary Table 5. Plasmid sequences were verified by DNA sequencing. TmFtn, as well as mutants, were purified as follows. First, the plasmids corresponding to FLAL and FLAL-L were transformed into BL21 (DE3) *Escherichia coli* cells, respectively, and then cultured at 37 °C in 1 L of LB media containing 100 μg/mL ampicillin. After the cell density reached an absorbance of 0.6 at 600 nm, protein expression was likewise induced with 200 μM IPTG for 10 h at 37 °C. Cells were harvested by centrifugation (8609 × *g*) and the precipitate was re-suspended in 50 mM Tris–HCl (pH 8.0), followed by sonication. The supernatant was collected from lysed cell samples and subjected to heating at 90 °C for 10 min. Then thermal-treated supernatant was collected after centrifugation at 8609 × *g* for 30 min and passed through a membrane filter. Finally, the protein solution was applied to an ion-exchange column (DEAE Sepharose Fast Flow, GE Healthcare), followed by gradient elution with 0–1.0 M NaCl. The purified protein was then dialyzed against 50 mM Tris–HCl (pH 8.0) at 4 °C to exclude NaCl from the solution, and protein concentrations were determined according to the Lowry method with bovine serum albumin as standard. Protein purity was confirmed by SDS–PAGE. The molecular weight of TmFtn subunit was estimated to be approximately 20 kDa, which is consistent with our electrophoretic band as shown in Supplementary Fig. 2b, and the molecular weight of the dimer is approximately 40 kDa.

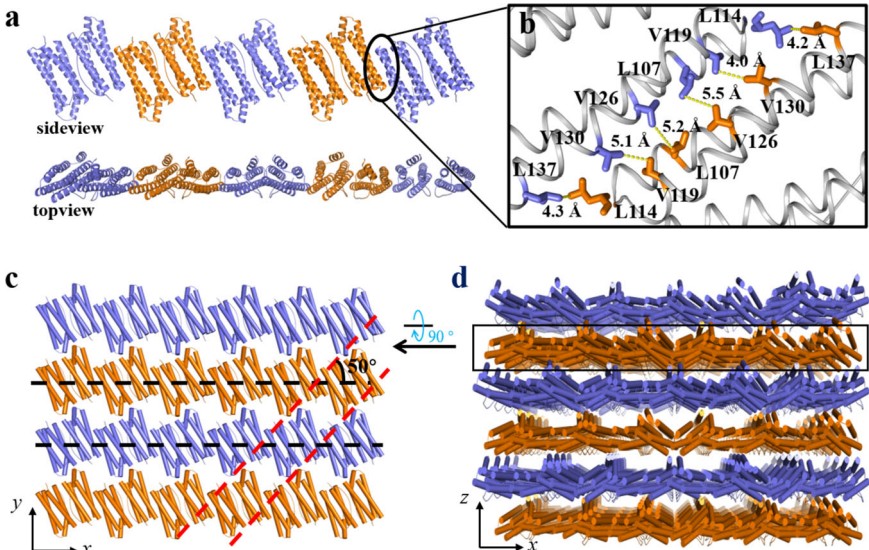

**Fig. 7 The crystal structure and arrangement of FLAL-L mutant. a** Side and top views of the 1D FLAL-L filament in the crystal structure. **b** Close-up view of the hydrophobic interactions between two adjacent FLAL-L molecules. **c** The 1D filaments coalesce in parallel along the *y* axes to form 2D protein layer, corresponding to the 2D ribbons observed by TEM. The dashed lines in **c** correspond to the lines shown in Fig. 6i. **d** From the top view, FLAL-L molecules assemble into 3D protein frameworks.

**Preparation of nanocages, filaments, nanorods, and nanoribbons**. After purification, TmFtn molecules remain as a dimer in the absence of metal ions. For protein nanocages, 50 mM Tris–HCl (pH 8.0) was used as buffer and 50 mM CaCl$_2$ was required to induce dimeric TmFtn self-assembly. Filaments were prepared by adding a certain amount of CaCl$_2$ to FLAL solution (48 μM) at room temperature. After being stirred for several minutes, the resulting mixture was incubated at 4 °C overnight. To optimize the assembly conditions, different concentrations of CaCl$_2$ were tested over a range of 0–100 mM. Similarly, nanorods were prepared by adding a certain amount of PEG1500 to FLAL solution (12 μM) at room temperature. After gently shaking for a few minutes, the resulting mixture was incubated at 4 °C overnight. Nanoribbons were prepared by mixing FLAL-L solution (12 μM) with 30% PEG1500, and then the reaction mixtures were incubated for 15 days. The concentrations and polymerization degree of PEG used were optimized.

**Docking simulations**. One thousand independent docking trajectories were carried out using RosettaDock[48]. The two TmFtn or FLAL subunits that form the designed interface were used as the starting structure for the docking simulations. One of the monomers was randomly spun along the axis connecting the centers of mass of both partners and the same monomer was also allowed to search a space of up to 3 Å normal to that axis, 8 Å in the plane perpendicular to the axis, and with up to an 8° tilt from the axis and an 8° additional spin around the axis.

**Polyacrylamide gel electrophoresis**. The purity and molecular weight of protein samples were estimated by PAGE. Gel electrophoresis under denaturing conditions was carried out using a 15% polyacrylamide–SDS gel as reported by Laemmli[56], and samples need to be heated in a water bath for 5 min. For native PAGE, a 4–20% polyacrylamide gradient gel was used and run at 5 mA for 10 h at 4 °C. Gels were stained with Coomassie brilliant blue R250.

**High-resolution gel filtration chromatography analyses**. High-resolution gel filtration chromatography analyses were performed using an ÄKTA pure system coupled to a Superdex 200 Increase column (GE Healthcare) in buffer (50 mM Tris, 100 mM NaCl, pH = 8.0) with a flow rate of 0.5 mL/min.

**TEM imaging**. Protein samples (10 μL) were deposited on carbon-coated copper grids and excess solution was removed with filter paper after a 2-min incubation. Then protein samples were stained using 2% uranyl acetate for 5 min. Transmission electron micrographs were obtained at 80 kV through a Hitachi H-7650 transmission electron microscope.

**AFM measurements**. For AFM sample preparation, protein samples (10 μL) were pipetted on freshly cleaved mica (Beijing Zhongxingbairui Technology Co., Ltd.) and dried at room temperature. The AFM images were collected using a Nanoman VS (Bruker) with tapping mode at a resolution of 256 lines per image and a scan rate of 1 Hz. The AFM images were processed using NanoScope Analysis.

**Crystallization, data collection, and structure determination**. Purified proteins were concentrated to 6 mg/mL in a buffer consisting of 20 mM Tris–HCl at pH 8.0, and crystals were obtained using the hanging drop vapor diffusion method under different conditions, which are shown in Supplementary Table 1. X-ray diffraction data were collected at Shanghai Synchrotron Radiation Facility (SSRF; BL17U and BL19U) with merging and scaling by the HKL-3000 software[57]. Data processing statistics are shown in Supplementary Table 2. The structures were determined by molecular replacement using the Molrep program in CCP4 using the structure of TmFtn (PDB code 1VLG) as a search model. Structure refinement was conducted using the Refmac5 program and PHENIX software[58]. The structure was rebuilt using COOT[59], which made the model manually adjusted. All figures of the resulting structures were produced using the PyMOL[60] program and UCSF[61] Chimera package.

## Data availability
The atomic coordinates and structure factors in this study have been deposited in the Protein Data Bank under the accession PDB IDs: 7DYA, 7DY8, 7DY9, and 7DYB. Source data are provided with this paper.

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

## Acknowledgements

This work was supported by the National Natural Science Foundation of China (Nos. 31972018 and 31730069). The Shanghai Synchrotron Radiation Facility (SSRF) is especially acknowledged for beam time. We thank the staff from BL17U1/BL18U1/19U1 beamline of the National Center for Protein Sciences Shanghai (NCPSS) at Shanghai Synchrotron Radiation Facility for assistance during data collection.

## Author contributions

G.Z. and T.Z. conceived and directed the project and wrote the paper. X.Z. designed and performed experiments, analyzed data, and co-wrote the paper. B.Z. performed experiments. Y.L. and H.W performed the crystal data collection. J.Z. and C.L. performed the experiments and co-wrote the paper.

## Competing interests

The authors declare no competing interests.
