## [Peer Review File · Nature Communications]

Reviewers' Comments:

Reviewer #1:

Remarks to the Author:

In this paper Zhang et al. have shown how *Thermotoga maritima* ferritin can be engineered such that in certain conditions it can form a linear chain or a ribbon, depending on conditions. They used a rational design approach and confirmed the formed structures by a range of biophysical techniques including a number of crystal structures.

Overall, the manuscript is written in an understandable way. The design of the protein interfaces gave a theoretically predicted outcome which is a success.

Minor Points

Lines 63-65, please consider revising; not clear what "inaccessible" means.

Line 147 – metal coordination is not providing any covalent interaction; this is purely ionic between metal ion and side chains, and coordination with water molecules

Lines 252 – 253; this is slightly unclear. Were the authors inspired by the structure of alpha keratin and decided to design a new controlled structure in this fashion or the structure happened to resemble alpha keratin by happenstance?

Line 201 – Asn is an amide, not an acidic residue

Line 231 – pi-pi interaction could/should be specified as "pi-stacking" since resolution of the obtained structures allows analysis at this level of details

I am curious as to how the authors came to the concentration of Ca^{2+} [80 mM], were other series of concentrations used? It would be good to provide any extra data or perhaps more supplementary data on the obtainment of this result.

The authors fail to comment about the potential applications of such novel nano-bio-structures, it would certainly contribute to improving the quality of the manuscript.

SuppFig8 – the most interesting assembly (panel c) was not explored further, which left a very big question open – how ferritin cages can assemble in 3D lattice and how the designed interface influences it

SuppFig9 – different magnification makes the figure difficult to read, especially panel a contains one assembled cage which

Major points

Nomenclature: Please revise your use of the word "conversion." This implies that the same structure can be converted into another structure simply by changing environmental conditions. This is not the case here different proteins (i.e. mutants) form different structures. Please change the nomenclature and make this clear. What the authors have done is in fact redesign which doesn't allow conversion upon stimuli. The only time it was shown that the ferritin cages assemble into 2D or even 3D arrays, would be a direct example of conversion from 0D to 1D/2D structures but this wasn't explored and analysed in detail.

Please revise the use of the terms 0D, 1D and 2D. These are misleading. The cage is not 0D it is a hollow 3D sphere made up of discrete 3D building blocks. The "3D" structure is actually a 3D lattice. I suggest using more accurate descriptors such as "hollow sphere", "nanowire" "ribbon" and "3D lattice"

Methods are insufficient e.g. how was PEG mixed and incubated to form nanorods. I think that methods need to be much more comprehensive to allow someone to fully replicate every experiment. A more comprehensive methods section should be included.

Please provide AFM analysis of all assemblies produced.

Please consider including High-res AFM analysis of cubic assembly made out of ferritin cages, which could provide direct evidence for the cage being a building block (suppFig8)

Supplementary Figure 9 should be repeated, particularly panel A which is quite blurry and unclear. In Supp Figs 3 and 12 it would be good to have more details of the interactions between side chains before and after Rosetta design.

Supplementary Table 3 shows different concentrations and species of PEG investigated for mutant FLAL-L, nonetheless, no such investigation was performed for mutant FLAL. The reviewer is curious

if there is any influence on the nano - bio - structure of that mutant.
English needs to be checked by a native speaker throughout as there are lots of awkward or difficult to follow phrases.

Reviewer #2

Remarks to the Author:

In this manuscript, the authors show their concept “protein interface redesign strategy” is able to transform dimer unit of 0D protein nanomaterials into different high-ordered architectures such as 1D filament, 1D nanorod and 2D nanoribbons. They chose dimer units of ferritin from *Thermotoga maritima* which self-assemble into 24-meric protein nanocage, responding to the additions of magnesium ions in nature. First, they show the nanocage self-assembly occurs with calcium ions instead of magnesium ions, and then re-design protein-protein interface using RosettaDock algorithm to assemble into different high-ordered structures. They used various different techniques to represent that these structures were made as they designed. Overall, their study is interesting and shows their re-design strategy works for this model study, but I have some concerns with its actual novelty as well as their design and data analysis through this manuscript.

Concern regarding the novelty based on previous work:

There is an existence of similar study in 2012 (reference 7 in this manuscript) to show the protein interface redesign to create metal mediated binding site between protein-protein interface of a single four-helical bundle protein. In this previous paper, they showed the formation of 1D, 2D and 3D protein assemblies which respond to metal coordination and/or pH changes. They also demonstrate the control of the morphology changes between different structures. This previous study showed similar concept of protein interface design, in which they called metal-directed protein assembly.

Concerns regarding the composition of this manuscript:

In fig. 1, authors should explain their design strategy rather than summary of this manuscript. Current fig. 1 is more likely conclusion or graphical abstract. I believe that main point of this manuscript is to introduce their new protein design approach. However, these concepts only describe briefly in text and show figures in supplemental information.

Second, their design (hydrophobic interactions between each protein unit) is not dependent on metal mediated assembly from natural protein. Therefore, it is not clear why authors need to feature this natural behavior in main figure (fig. 2). At the end, the addition of metal is needed to

assembly 1D filament, but this should not be the main part for this manuscript.

In addition, it is not clear why authors need to demonstrate calcium ion mediated assembly, not just using magnesium ions in natural form. Again, this does not seem relevant to their concept for this manuscript. It is better to explain if it's important for their concept in this paper.

Concerns regarding their design and data through this manuscript:

Authors mention that their design strategy is "simple and effective approach in abstract (line 17)" and "straightforward strategy in main text (line 66)". However, their design itself was not able to create their expected protein assembly. Indeed, they needed many trial-and-errors to tune fine conditions to form high-ordered structures with addition of the external factors such as calcium ion and detergent, which was not in their design. Therefore, it is hard to agree that their design is simple and effective approach since it did not work as they expected. In addition, it seems that their design and findings are limited to this model protein. There are also many concerns and questions regarding their design, experiments, and data analysis as shown below.

Additional comments for their experiments and analysis in this manuscript:

1. Title misleads the content of manuscript (also line 17 and 22). First, I thought authors were able to convert from 0D nanocage structure into 1D and 2D, not dimer unit of 0D nanocage structure.
2. In line 141, they mentioned that several acidic amino acids prevent the formation of their expected structure. In supplementary fig. 7, they indicated 6 acidic amino acids, and there are additional possible 3 acidic amino acids in C-helix (another helix involved in proteinprotein interface). Wasn't it able to detect while processing three step computational method? It seems too many of them to start with.
3. In line 150, authors suggested that calcium ion and protein concentration is closely associated for the formation of protein assembly. According to fig. 4, the ratio of protein and calcium ion should be 1:1 to form 1D filament, but in experiments, authors used ratio of about 1: 1600. Why does ~1600 times excess amount of calcium ion need to form 1D structure? In addition, authors observed two kind of protein array. Is it possible to control the shape of protein arrays one to another, using different ratio of protein to calcium as of ref 7 (Nat. Chem. 4, 375-382 (2012))? Also, author might be able to get additional insight from paper, Nat. Commun.10, 5545 (2019). This paper should be also good to cite.
4. In line 170, authors indicate "these finding approving our design", however, their design

is supposed to form 1D filament without calcium ion.

5. In line 175, authors suggested the longest length of the 1D filament is 1 micro meter from fig. 3d. It might be right, but it is not able to see 1 micro meter of filament from figure.

Authors also mentioned in line 178 that “usually two or four filaments stack together”, but it seems that there are many other different stackings observed in fig. 3. If authors have specific reasons to select these two components, please indicate more clear data and their analysis. Also, why does only one side (2 nm in width according to supp. fig. 1) observe in TEM images, not the others (5nm in length)?

6. In line 207, authors mention the electrostatic interaction of filament each other due to Lys 10 and Glu164. To create 1D filaments, it seems authors should make mutant protein which do not contain these two amino acids. At this moment, it is not distinguished 1D filaments observed as author described in text.

7. From line 210, authors describe the addition of PEG to create different protein assembly (1D nanorod), but this part seems off the point since their main concept of this manuscript is to show “a simple and effective protein interface redesign strategy”, not depending on external factors.

8. Line 217 – 224 and fig. 5: It is hard to understand what authors try to show and confirm in fig. 5, especially fig 5. d. I assumed that it is reconstructed image from Fourier transforms of Fig. 5.c; however, it is not clear data to show the structure as authors indicated in line 222 “an excellent view of the structure of the 1 D nanorod”. If authors want to discuss this data, they need to add scale and explain lattice pattern more in detail. In addition, what is yellow lines in fig. 5(b) ?

9. In line 234, authors mentioned “confirming our design in atomic detail”. What kind of design do authors confirm in here?

10. In line 301, authors indicate “The crystal structure provide unambiguous evidence to confirm our design in atomic detail”. Please clear what your design is indicated here.

Minor comments:

1. Line 111: change word “salts” to “metals (or more relevant word)”
2. Line 142: change “several” to “many” as you mentioned in supp fig. 7
3. Line 154: change liner-shaped protein arrays to 1D filaments
4. Line 127: add reference 53 for RossetaDock algorithm here

5. Line 380, 200 mM IPTG? Is it microM?
6. Line 383, 387, 402, fix open square to Celsius
7. Fig. 3 (e) caption: high "magnification", not "resolution"
8. Fig. 3 (d) and supp fig. 8(d): please check if scale is correct. Size of materials (or magnification) look different.
9. Supplementary fig. 3: please mention protein size in caption
10. Supplementary fig. 8 (d): red arrows, not blue.

Reviewer #3:

Remarks to the Author:

Zhang et al. describe the design and structural characterization of a number of variants of *T. maritima* ferritin that assemble to various architectures. *T. maritima* ferritin appears to be somewhat unique in that it only assembles to a dimer (not further) in the absence of divalent metal ions, whereas most ferritins assemble (at least partially) to the octahedral 24mer. The authors confirm that *T. maritima* ferritin assembly beyond the dimer requires metal ions, and further redesign the intermolecular interface of *T. maritima* ferritin to drive assembly of a number of different architectures. The work overall is of high technical quality and the conclusions are well-supported by the data. It is a nice paper that warrants publication in Nature Communications.

::Major comments

-- I don't have any major hesitations with the work being published. The data and analyses are of high quality. The results are described in enough detail, and clarified in the figures, that the reader will be able to readily ascertain the import of the results.

::Minor comments

-- Lines 235-331: The following passage requires revision: "However, how to convert zero-dimensional protein cage into one- or two-dimensional architecture by a simple, effective method in the laboratory has yet to be explored. The challenge lies in the fact that the construction of 0D protein nanocage belongs to intramolecular assembly at the protein quaternary level, while the formation of the 1D or 2D nanomaterials by using the same building blocks are mainly associated with intermolecular assembly. Therefore, switching the intramolecular assembly of protein building blocks to its intermolecular analogue is the key to realize 0D→1D nanomaterials." It is not true that assembly of 0D protein nanocages relies only on "intramolecular assembly" -- this is non-sensical, as even the formation of nanocages obviously relies on "intermolecular assembly". The key is the geometry of the protein-protein interactions that drive assembly. In certain arrangements, these interactions drive the assembly of 0D materials, while in other arrangements, they drive the assembly of other materials.

Response to comments from reviewer 1:

1) Lines 63-65, please consider revising; not clear what “inaccessible” means.

Response: Suggestion was followed in the revised manuscript. In fact, although many efforts have been made to construct various self-assembly architectures, but to the best of our knowledge, controlling the transformation of protein nanocages into 1D or 2D nanomaterials by regulating spatial rearrangement of similar protein building blocks remains unexplored. In the revised version, we have changed the sentence to “Despite these advances, rendering PPIs controllable to facilitate the transformation of hollow protein nanocage into 1D or high order nanomaterials with similar building blocks in the laboratory has yet to be explored”.

2) Line 147-metal coordination is not providing any covalent interaction; this is purely ionic between metal ion and sidechains, and coordination with water molecules.

Response: Suggestion was followed. To avoid the misunderstanding, we replaced the “coordination” with “interaction” in the revised manuscript.

3) Lines 252 - 253; this is slightly unclear. Were the authors inspired by the structure of alpha keratin and decided to design a new controlled structure in this fashion or the structure happened to resemble alpha keratin by happenstance?

Response: Suggestion was followed in the revised manuscript, and we tried to make this clear. Inspired by the packing pattern of α -keratin protofilaments where two keratin polypeptides first form a dimeric coiled coil, and then such formed coiled coils as building blocks are staggered with each other in a head to tail manner, so we decide to adjust the fully side-by-side interaction manner of two adjacent dimeric protein building blocks into a partially side-by-side pattern, namely a staggered manner.

4) Line 201 – Ans is an amide, not an acidic residue

Response: Suggestion was followed, and correction was made in the revised manuscript.

5) Line 231 – pi-pi interaction could/should be specified as “pi -stacking” since resolution of the obtained structures allows analysis at this level of details

Response: Suggestion was followed, and corrections were made in the revised manuscript.

6) I am curious as to how the authors came to the concentration of Ca^{2+} [80 mM], were other series of concentrations used? It would be good to provide any extra data or perhaps more supplementary data on the obtainment of this result.

Response: Suggestion was followed. Actually, we did a series of optimization against protein and Ca^{2+} concentration, and results showed that the filaments appeared at higher concentration of Ca^{2+} and FLAL. Considering the behaviors derived from different combinations between protein and Ca^{2+} , we deemed the optimal condition for the formation of filaments is $[\text{FLAL}] = 48.0 \mu\text{M}$ and $[\text{Ca}^{2+}] = 80.0 \text{mM}$. This information has been added as Supplementary Table 3 in the revised manuscript.

Supplementary Table 3 Behaviors of FLAL mutant at different concentrations of FLAL and Ca^{2+} .

		Ca^{2+} concentration				
		20 mM	40 mM	60 mM	80 mM	100 mM
Protein concentration	12 μM	N	N	N	N	A
	24 μM	N	N	N	N	A
	36 μM	A	A	A	F	F
	48 μM	A	A	F	F	F
	60 μM	A	A	F	F	F

N - No assembly; A - Aggregation; F - Filament.

7) The authors fail to comment about the potential applications of such novel nano - bio - structures, it would certainly contribute to improving the quality of the manuscript.

Response: We appreciate this suggestion. Comment about the potential applications of such novel structures was added in the Discussion section. One literature related to the gradated assembly of multiple different expressed proteins was added to the revised version as reference 55:

55. Hudalla, G. A. et al. Gradated assembly of multiple proteins into supramolecular nanomaterials.

Nat. Mater. **13**, 829-836 (2014).

8) SuppFig8 – the most interesting assembly (panel c) was not explored further, which left a very big question open – how ferritin cages can assemble in 3D lattice and how the designed interface influences it.

Response: Thanks for your suggestion. In our previous study (reference 16 in this manuscript), we have used ferritin nanocage as building blocks to fabricate 2D and 3D superlattices. By introducing natural aromatic amino acid residues such as Phe, Tyr, and Trp into the C_4 interface of ferritin, ferritin nanocages self-assemble into 3D lattices through the π - π interactions. Subsequently, we have fabricated similar superlattices utilizing hydrophobic interactions (reference 17 in this manuscript) and metal coordination (reference 19 in this manuscript). In this study, Phe147 located at the C_4 interface of 24-meric protein nanocage, and all the genetic modification hardly affecting the inherent assembly property. Based on this, when the Ca^{2+} was added in the protein solution, dimeric FLAL firstly assembled into protein nanocage and then assembled into 3D superlattices through π - π interactions. Since similar phenomenon has been published by our group, we do not elaborate much on it in this manuscript.

9) SuppFig9 – different magnification makes the figure difficult to read, especially panel a contains one assembled cage which.

Response: Suggestion was followed in the revised version. This experiment was repeated, and new TEM images of TmFtn in the absence and presence of 80 mM Ca^{2+} were added to substitute the previous one in the revised manuscript as Supplementary Figure 10.

10) Nomenclature: Please revise your use of the word “conversion.” This implies that the same structure can be converted into another structure simply by changing environmental conditions. This is not the case here different proteins (i.e. mutants) form different structures. Please change the nomenclature and make this clear. What the authors have done is in fact redesign which doesn’t allow conversion upon stimuli. The only time it was shown that the ferritin cages assembly into 2D or even 3D arrays, would be a direct example of conversion from 0D to 1D/2D structures but this wasn’t explored and analysed in detail.

Response: Thanks for your suggestion. And your suggestion was followed in the revised version. The word “conversion” was eliminated in the revised manuscript, except for the conversion between

nanorods to nanoribbons.

11) Please revise the use of the terms 0D, 1D and 2D. These are misleading. The cage is not 0D it is a hollow 3D sphere made up of discrete 3D building blocks. The “3D” structure is actually a 3D lattice. I suggest using more accurate descriptors such as “hollow sphere”, “nanowire” “ribbon” and “3D lattice”

Response: Suggestion was followed. “Zero-dimensional protein nanocage” has been changed to “hollow protein nanocage”, and “filament”, “nanoribbon” and “3D superlattice” were used to describe the assembled structure in the revised version.

12) Methods are insufficient e.g. hoe was PEG mixed an incubated to form nanorods. I think that methods need to be much more comprehensive to allow someone to fully replicate every experiment. A more comprehensive methods section should be included.

Response: Suggestion was followed. In the revised version, more detailed information related to the experimental procedure was added in the part of Methods, especially the preparation of hollow protein nanocages, filaments, nanorods and nanoribbons.

13) Please provide AFM analysis of all assemblies produced.

Response: We have tried to use AFM to analysis all the assemblies produced in our work, and the AFM analysis of the nanorod was added as Supplementary Figure 14, which revealed that the height of the nanorod is 220 ± 5 nm. However, the 48 μM FLAL and 80 mM Ca^{2+} resulting product was a mixture and thus we failed to capture a good AFM image.

Supplementary Figure 14. Characterization of FLAL nanorod by atomic force microscopy

(AFM). (a) AFM image of the nanorod. (b) Height distributions measured along the white two-way arrows in (a).

14) Please consider including High-res AFM analysis of cubic assembly made out of ferritin cages, which could provide direct evidence for the cage being a building block (supplFig8)

Response: Suggestion was followed. To provide a direct evidence for protein cages as building blocks, we added a new TEM image as Supplementary Figure 9c in the revised manuscript instead of high-resolution AFM, which confirmed that the cubic assembly is generated with ferritin cages as building blocks.

Supplementary Figure 9c. Magnified view of the cubic assembly obtained by adding 80 mM Ca^{2+} to 48 μM FLAL.

15) Supplementary Figure 9 should be repeated, particularly panel A which is quite blurry and unclear.

Response: Suggestion was followed. We have repeated this experiment, and clear TEM images of TmFtn in the absence and presence of 80 mM Ca^{2+} were used to substitute the previous figure in the revised manuscript as Supplementary Figure 10.

Supplementary Figure 10. TEM images of TmFtn in the absence (a) and presence of 80 mM Ca^{2+} (b). Conditions: [TmFtn] = 48.0 μM in 50 mM Tris-HCl, pH 8.0, [Ca^{2+}] = 80 mM. (a, b) Scale bars represent 100 nm.

16) In Supp Figs 3 and 12 it would be good to have more details of the interactions between side chains before and after Rosetta design.

Response: Thanks for your suggestion. The dimeric TmFtn molecules self-assemble in a head-to-side manner to form protein nanocages, but Rosetta design was carried out to give a suitable model of which the dimeric molecules interact side-by-side. Thus, the interactions come from the amino acid located at the C_3 - C_4 interface when dimers interact head-to-side (Cover letter Figs. 1a-c), while the amino acids from C and D helices of each dimer are responsible for the interaction when dimers interact side-by-side (Cover letter Fig. 1d, e). And the designed interactions are prerequisites for the side-by-side interaction manner. However, we did the Rosetta design without side-chain sampling, which means no mutation was made, thus the designed interaction did not show in the Rosetta-calculated model, but we show where they supposed to be in the cover letter Fig. 1e. In our work, we tuned the interaction manner of dimeric building block from a head-to-side pattern to a side-by-side pattern without focus on replace the existed interaction in the C_3 - C_4 interface with designed interactions at the D helix to drive the dimer molecules interact side-by-side. Thus, we did not give a detailed description of the interactions between side chains before and after Rosetta design in the manuscript. However, we supplemented a structure comparison between the designed structures and solved crystal structures as Supplementary Figure 20 in the manuscript.

Cover letter Fig. 1. The interactions between side chains before and after Rosetta design. (a) The dimeric TmFtn molecules self-assemble in a head-to-side manner to form protein nanocage. (b, c) The interactions between side chains when dimers interact head-to-side. (d) The Rosetta-calculated model of which the dimeric molecules interact side-by-side. (e) The interactions between side chains when dimers interact side-by-side, but without the designed interactions.

17) Supplementary Table 3 shows different concentrations and species of PEG investigated for mutant FLAL-L, nonetheless, no such investigation was performed for mutant FLAL. The reviewer is curious if there is any influence on the nano - bio - structure of that mutant.

Response: Suggestion was followed in the revised version. For mutant FLAL, the concentrations and species of PEG effects on the formation of nanorod were optimized. After trying PEG1000, PEG1500 and PEG3000 with concentrations of 10%, 20%, 30%, the nanorod existed only in the presence of 30% PEG with an average molecular weight of 1500 (PEG1500). This information was added in the revised manuscript.

18) English needs to be checked by a native speaker throughout as there are lots of awkward or difficult to follow phrases.

Response: Suggestion was followed. A native speaker has checked the revised manuscript

throughout, and corresponding corrections were made in the revised version.

Response to comments from reviewer 2:

1) There is an existence of similar study in 2012 (reference 7 in this manuscript) to show the protein interface redesign to create metal mediated binding site between protein-protein interface of a single four-helical bundle protein. In this previous paper, they showed the formation of 1D, 2D and 3D protein assemblies which response to metal coordination and/or pH changes. They also demonstrate the control of the morphology changes between different structures. This previous study showed similar concept of protein interface design, in which they called metal-directed protein assembly.

Response: I agree with you that the reference 7 work from Dr. Tezcan group was a milestone job in the area of protein design and assembly. But our research work presented in this manuscript is different from the work reported by reference 7, and we could claim our novelty from three aspects as bellow. Firstly, our protein design strategy is different from that by Dr. Tezcan group. In this manuscript, we highlight a new concept that the transformation of protein nanocage into 1D or 2D protein nanomaterials can be realized by regulating the spatial re-arrangement of protein building blocks, namely a head-to-side manner *versus* a side-to-side manner. Secondly, in the reference 7, Tezcan group utilize metal-directed, chemically tunable assembly and claim that ‘The premise of our approach is that metal-bonding interactions can capture all the salient features of protein-protein interactions (stability, specificity, directionality, symmetry and reversibility) on a much smaller surface than is needed by non-covalent interactions, and thus require ‘little design’’. However the protein-protein interactions that we designed in this manuscript are mainly derived from noncovalent interactions (π - π interactions and hydrophobic interactions), which are much common in the natural protein interactions. Thirdly, reference 7 summarized ‘‘Previous efforts in designed protein self-assembly had limited success: generally, they required building blocks with high pre-existing symmetry, allowed little or no external control over self-assembly and yielded static protein architectures that did not display long-range order beyond a few hundred nanometers’’. In this manuscript we did not utilize the pre-existing symmetry, instead we redesign the dimeric protein interface to break the pre-existing symmetry, leading to a totally different assembly into micrometers architectures with external stimulation. Based on these differences we claim our novelty different

from the reference 7 and complement some discussion over reference 7 in the revised manuscript.

2) In fig. 1, authors should explain their design strategy rather than summary of this manuscript. Current fig. 1 is more likely conclusion or graphical abstract. I believe that main point of this manuscript is to introduce their new protein design approach. However, these concepts only describe briefly in text and show figures in supplemental information.

Second, their design (hydrophobic interactions between each protein unit) is not dependent on metal mediated assembly from natural protein. Therefore, it is not clear why authors need to feature this natural behavior in main figure (fig. 2). At the end, the addition of metal is needed to assembly 1D filament, but this should not be the main part for this manuscript.

In addition, it is not clear why authors need to demonstrate calcium ion mediated assembly, not just using magnesium ions in natural form. Again, this does not seem relevant to their concept for this manuscript. It is better to explain if it's important for their concept in this paper.

Response: Thanks for your comments. Indeed, the main point of this manuscript is to introduce our new protein design approach, and this approach can be summarized as tuning the interaction manner of protein building blocks. Besides, the purpose of this design approach is to transform protein nanocage into 1D and 2D assemblies. In the revised Figure 1, the left column of the figure shows how the building blocks interact, from the initial head-to-side manner to a fully side-by-side, and then to a partially side-by-side manner, which is the design concept of this research. Among them, the head-to-side manner represents the natural tendency of this building block interaction, while the fully and partially side-by-side manner is realized through protein interface redesign. Thus, the right column of the figure confirms that our protein design approach is feasible and achieved the transformation from protein nanocage to filaments, nanorods and nanoribbons. In all, the current Figure 1 describes our design approach and confirms its feasibility.

Obviously, our design is not dependent on metal mediated assembly from natural protein, but we believe that showing the natural behavior of the building block is important. The reasons are as follows: (a) In this research, we transformed hollow protein nanocage into 1D and 2D assemblies, so the natural behavior of the building block is the starting point of our experiment; (b) the formed hollow protein nanocage provides the information that adjacent dimeric molecules interact in a head-to-side manner, which inspires us to control the assembly by adjusting the interaction manner

of the building blocks. Based on these considerations, we describe this natural feature in the manuscript. To make our concept focus on the transformation, we **followed your suggestion** and put this figure in Supporting Information as Supplementary Figure 3 in the revised manuscript. The addition of the calcium ion is not the main part but part of the design concept, the interaction generated by calcium coordination was important to strength the designed side-by-side interaction to realize the 1D filament assembly.

In addition, it has been reported that divalent cations are more effective in promoting assembly, and we confirmed that both divalent calcium and magnesium ions can mediate the assembly of dimeric TmFtn into nanocage. However, magnesium was not able to play the same role as calcium to strength the side-by-side building blocks interaction and transform into linear assembly (**Cover letter Fig. 2**). Thus, we kept using calcium all through this manuscript. Again, the concept was to redesign the building blocks interact interface and realize the different higher dimensional nanomaterials from similar dimeric building block, and the external stimulation was assistant approach to adjust the interaction strength.

According to your suggestion, we have made corrections in the revised manuscript.

Cover letter Fig. 2. Characterization of the assembly state after adding 80 mM Mg^{2+} in 48.0 μM FLAL solution for 24 h. Only protein nanocage can be observed under TEM.

3) Authors mention that their design strategy is “simple and effective approach in abstract (line 17)” and “straightforward strategy in main text (line 66)”. However, their design itself was not able to create their expected protein assembly. Indeed, they needed many trial-and-errors to tune conditions to form high-ordered structures with addition of the external factors such as calcium ion and detergent, which was not in their design. Therefore, it is hard to agree that their design is simple

and effective approach since it did not work as they expected. In addition, it seems that their design and findings are limited to this model protein. There are also many concerns and questions regarding their design, experiments, and data analysis as shown below.

Response: Thanks for your comments. Just as Tezcan claimed in the reference 7: “the designed self-assembly of proteins has largely been inaccessible, plagued by the chemical heterogeneity and size of the molecular surfaces required for protein-protein interactions”, we all know that protein self-assembly design is not easily compared with DNA, RNA and peptides. The “simple and effective” were relatively speaking compared with the *de novo* design from Dr. David Baker and other strategies of protein assembly design. Our design concept was just tuning the interaction manner of two adjacent protein building blocks to switch protein higher dimensional ordered stacking. Along the designed interactions the metal ion coordination as well as direct site mutations are all the realizing approaches. Our reported crystal structures demonstrate that the designed non-covalent interactions indeed play a key role in the formation of the filament, nanorod, and ribbon. However, in solution, the designed interactions are not strong enough, so external stimuli were needed to trigger the generation of these 1D and 2D materials. Consistent with our design, natural dimeric TmFtn cannot form such 1D and 2D nanomaterials in the presence of external stimuli such as calcium ions and PEG both in solution and protein crystals.

4) Title misleads the content of manuscript (also line 17 and 22). First, I thought authors were able to convert from 0D nanocage structure into 1D and 2D, not dimer unit of 0D nanocage structure.

Response: Suggestion was followed. The title was revised into “Protein interface redesign facilitates the transformation of hollow protein nanocages into 1D or 2D nanomaterials with similar dimeric building blocks” in the revised manuscript.

5) In line 141, they mentioned that several acidic amino acids prevent the formation of their expected structure. In supplementary fig. 7, they indicated 6 acidic amino acids, and there are additional possible 3 acidic amino acids in C-helix (another helix involved in protein-protein interface). Wasn't it able to detect while processing three step computational method? It seems too many of them to start with.

Response: Suggestions was followed. In the three-step computational assisted method,

RosettaDock algorithm was used to give a suitable calculated model of which dimeric TmFtn molecules interact side-by-side. The installation of new interactions and the analysis of amino acid residues at the new interface were not processed with Rosetta. There are many acidic amino acids at the new interface, and D-helix is in dominate, so we mainly show the acidic amino acids in D-helix. According to your suggestion, we have made corrections in the revised manuscript as shown in Supplementary Figure 8.

6) In line 150, authors suggested that calcium ion and protein concentration is closely associated for the formation of protein assembly. According to fig. 4, the ratio of protein and calcium ion should be 1:1 to form 1D filament, but in experiments, authors used ratio of about 1: 1600. Why does ~1600 times excess amount of calcium ion need to form 1D structure? In addition, authors observed two kind of protein array. Is it possible to control the shape of protein arrays one to another, using different ratio of protein to calcium as of ref 7 (Nat. Chem. 4, 375-382 (2012))? Also, author might be able to get additional insight from paper, Nat. Commun.10, 5545 (2019). This paper should be also good to cite.

Response: Suggestions were followed. For the formation of filament, the concentration of Ca^{2+} and FLAL was optimized as shown in Supplementary Table 3 in the revised manuscript. Results showing that the optimal condition for the formation of filaments is $[\text{FLAL}] = 48.0 \mu\text{M}$ and $[\text{Ca}^{2+}] = 80.0 \text{ mM}$. On one hand, it has been reported (reference 46 in this manuscript) that 50 mM MgCl_2 or CaCl_2 is required to induce the self-assembly of dimeric TmFtn into protein nanocage, which represent the low affinity of calcium ions toward TmFtn. On the other hand, according to the crystal structure, the ratio of protein and calcium ion should be 1:1 to form filament, to be noted that this ratio was bounded calcium ion with obvious densities in the crystal structure, the weak bounded, nonspecific binding, or electronic interactions were not counted. Considering the ion equilibration and relative weak binding in the solution, 1: 1600 ratio was reasonable.

Our design strategy was generating different interactions from different subunit interfaces to realize different subunits assembly, the calcium coordination geometry was not considered here, so we were not able to control the shape of protein array with different calcium concentration.

Additionally, thanks for your suggestion, the paper, Nat. Commun.10, 5545 (2019), was added as reference 20 in the revised manuscript.

7) In line 170, authors indicate “these finding approving our design”, however, their design is supposed to form 1D filament without calcium ion.

Response: Thanks for this comment, and we respond this comment on the question 3, and corresponding modifications were made in the revised manuscript.

8) In line 175, authors suggested the longest length of the 1D filament is 1 micro meter from fig. 3d. It might be right, but it is not able to see 1 micro meter of filament from figure. Authors also mentioned in line 178 that “usually two or four filaments stack together”, but it seems that there are many other different stackings observed in fig. 3. If authors have specific reasons to select these two components, please indicate more clear data and their analysis. Also, why does only one side (2 nm in width according to supp. fig. 1) observe in TEM images, not the others (5nm in length)?

Response: Suggestions were followed in the revised manuscript. Firstly, we added a new result as Supplementary Figure 11 to the revised manuscript, showing that the length of the filaments can grow up to 1.3 μm .

Supplementary Figure 11. (a) TEM images of filaments constructed by 48.0 μM FLAL and 80 mM Ca^{2+} , and the length of the filament can reach 1.3 μm . (b) Enlargement of the filament in panel a.

Secondly, Figure 3 (now is figure 2 in the revised manuscript) shows the formation process of filaments. For short-time reaction or at low concentrations of filaments (especially Figure 2e, f), it is clear that two or four filaments together are usually observed under TEM. As times go on, the

filaments grow longer and more numerous, and at 24h, the large amounts of filaments are tangled together as shown in Figure 2d. Even though, they were still be recognized as two or four filaments.

Thirdly, according to the crystal structure, the stacked filaments are parallel-displaced from the side view as shown in revised Supplementary Figure 12b, and such stacking pattern may be responsible for the fact that only one side (2 nm in width) can be observed in TEM images. As shown in cover letter Fig. 3, the position associated with the 2 nm side is more stable on carbon-coated copper grids compared with the 5 nm side.

Cover letter Fig. 3 Schematic diagram of the situation when the filament is on carbon-coated copper grids. (a) Top view of the 2 nm side of filaments. (b) Top view of the 5 nm side of filaments. The blue arrows indicate the extension direction of assembly.

9) In line 207, authors mention the electrostatic interaction of filament each other due to Lys 10 and Glu164. To create 1D filaments, it seems authors should make mutant protein which do not contain these two amino acids. At this moment, it is not distinguished 1D filaments observed as author described in text.

Response: Suggestion was followed in the revised version. We prepared a mutant named M1 where Lys10 and Glu164 were replaced by Gly to test if the electrostatic interaction between Lys10 and Glu164 is critical for the stacking of filaments. TEM characterization of the formed filaments (Supplementary Fig. 13) showed that they were similar to the FLAL filaments, suggesting that the electrostatic interaction is not the only interaction, and other kinds of non-covalent interactions also occur in the filament. However, other interactions can't be distinguished under TEM, and we failed to solve the M1 crystal structure. Based on these results and your suggestion, we have made correction in the revised manuscript.

Supplementary Fig. 13. SDS-PAGE (a) and Native-PAGE (b) analyses of purified M1. Lane M, protein markers and their corresponding molecular masses. (c) TEM images of filaments constructed by 48.0 μM M1 and 80 mM Ca^{2+} . The scale bar represents 100 nm.

10) From line 210, authors describe the addition of PEG to create different protein assembly (1D nanorod), but this part seems off the point since their main concept of this manuscript is to show “a simple and effective protein interface redesign strategy”, not depending on external factors.

Response: Thanks for this comment again, and we addressed this question in the comments 2 and 3. The calcium and PEG are in the same situation, and we made corresponding modifications all through the revised manuscript.

11) Line 217 – 224 and fig. 5: It is hard to understand what authors try to show and confirm in fig. 5, especially fig 5. d. I assumed that it is reconstructed image from Fourier transforms of Fig. 5.c; however, it is not clear data to show the structure as authors indicated in line 222 “an excellent view of the structure of the 1 D nanorod”. If authors want to discuss this data, they need to add scale and explain lattice pattern more in detail. In addition, what is yellow lines in fig. 5(b)?

Response: Suggestions were followed. Figure 5, which is Figure 4 in the revised manuscript, is intended to show TEM analyses of the nanorods constructed by 12.0 μM FLAL and 30 % PEG 1500. Figure 4c is real map of the inverted FFT from panel b, giving a clear view of the nanorods. This result is preliminary revealed that the head-to-side interaction manner was destroyed in this situation. But the detailed structure information needs to be observed through crystal diffraction. Besides, the yellow line indicates the linear stacking in this structure, which is not necessary and we have removed it. We have made corresponding corrections in the revised manuscript.

12) In line 234, authors mentioned “confirming our design in atomic detail”. What kind of design do authors confirm in here?

Response: Thanks for this comment. Our design goal is to transform hollow protein nanocages into 1D and 2D assemblies through adjusting the interaction manner of dimeric building blocks. The crystal structure provides direct evidence that the dimeric FLAL molecules interact side-by-side to form filaments dependent on the designed π - π stacking interactions and hydrophobic interaction. Such packing pattern is obviously different from the head-to-side manner which is observed in the crystal structure of naturally dimeric TmFtn molecules, which is used for the formation of protein nanocage. Thus, the crystal structure confirms our design. Further support for our design comes from the fact that that natural dimeric TmFtn cannot form such 1D and 2D nanomaterials in the presence of external stimuli in solution and protein crystals.

13) In line 301, authors indicate “The crystal structure provide unambiguous evidence to confirm our design in atomic detail”. Please clear what your design is indicated here.

Response: Thanks for this comment. Again, our design goal is to transform hollow protein nanocages into 1D and 2D assemblies through adjusting the interaction manner of building blocks. Here, the crystal structure clearly shows that FLAL-L molecules are aligned with each other in a partially side-by-side manner to form filaments and even nanoribbons rather than protein nanocage. Thus, the crystal structure confirms our design. Further support for our design comes from the fact that that natural dimeric TmFtn cannot form such 1D and 2D nanomaterials in the presence of external stimuli in solution and protein crystals.

14) Line 111: change word “salts” to “metals (or more relevant word)”

Response: Suggestions were followed in the revised version.

15) Line 142: change “several” to “many” as you mentioned in supp fig. 7

Response: Suggestion was followed in the revised version.

16) Line 154: change liner-shaped protein arrays to 1D filaments

Response: Suggestion was followed in the revised version.

17) Line 127: add reference 53 for RosettaDock algorithm here.

Response: Suggestion was followed in the revised version.

18) Line 380, 200 mM IPTG? Is it microM?

Response: Thanks for this comment. This was a mistake, and it should be “200 μ M IPTG”, and we have made correction in the revised manuscript.

19) Line 383, 387, 402, fix open square to Celsius

Response: Suggestion was followed in the revised version.

20) Fig. 3 (e) caption: high “magnification”, not “resolution”

Response: Suggestion was followed in the revised version.

21) Fig. 3 (d) and supp fig. 8(d): please check if scale is correct. Size of materials (or magnification) look different.

Response: Thanks for the comment. The scale is correct. Figure 3d and Supplementary Figure 8d (now is Figure 2d and Supplementary Figure 9e in the revised manuscript) show filaments formed at different time scale. The filaments in Figure 2d are formed after 24 hours of reaction, while the filaments in Supplementary Figure 9e are formed after 30 min of reaction. Thus, they are different in size. We have added this information in the revised manuscript.

22) Supplementary fig. 3: please mention protein size in caption.

Response: Suggestion was followed in the revised version.

23) Supplementary fig. 8 (d): red arrows, not blue

Response: Suggestion was followed in the revised version.

Response to comments from reviewer 3:

1) Lines 235-331: The following passage requires revision: "However, how to convert zero-dimensional protein cage into one- or two-dimensional architecture by a simple, effective method in the laboratory has yet to be explored. The challenge lies in the fact that the construction of 0D protein nanocage belongs to intramolecular assembly at the protein quaternary level, while the formation of the 1D or 2D nanomaterials by using the same building blocks are mainly associated with intermolecular assembly. Therefore, switching the intramolecular assembly of protein building blocks to its intermolecular analogue is the key to realize 0D→1D nanomaterials." It is not true that assembly of 0D protein nanocages relies only on "intramolecular assembly" -- this is non-sensical, as even the formation of nanocages obviously relies on "intermolecular assembly". The key is the geometry of the protein-protein interactions that drive assembly. In certain arrangements, these interactions drive the assembly of 0D materials, while in other arrangements, they drive the assembly of other materials.

Response: Suggestion was followed. Based on your suggestion and other reviewer's comments, we have made corresponding corrections in the revised manuscript.

After making these changes and responding to the queries in the individual reviews, we feel that our paper now meets the standards of *Nature Communications*. Thank you for your further consideration. We look forward to hearing from you.

With kind regards,

Sincerely,

Guanghua Zhao

Reviewers' Comments:

Reviewer #1:

Remarks to the Author:

The authors have been thorough in addressing the points raised. I think the work is now acceptable for publication with one proviso: The English of the newly added text is in parts quite poor and really does need rectifying prior to publication.

Reviewer #2:

Remarks to the Author:

Revised manuscript is definitely improved, but still remains some concerns - same points as of previous comments.

1. Title: as of previous comment, it is still misleading.

It should be something like "Protein interface redesign facilitates the transformation of hollow protein nanocages' building block to assemble 1D and 2D nanomaterials" or "Protein building block interface redesign of hollow nanocage transforms the assembly of 1D and 2D nanomaterials".

Same for line 62, 79,169, 328. Again, readers will think that you transform "from cage into another high ordered structures". You converted (or transformed or redesigned) a building block of hollow protein nanocage to be able to form different high ordered structures. Indeed, line 248 makes this part very clear.

2. In addition to comment 1, in Fig 1., you should remove " β Transformation" to avoid misleading as well - Cage do not transform into 1D structure.

3. I still think you better to add scheme (figure) of "protein interface redesign", not just one showing very brief idea in left part of Fig. 1 since your concept of this manuscript is protein interface redesign. I think this addition will help readers to understand manuscript much smoother. (Same concern as 1st paragraph of concerns regarding the composition of this manuscript)

4. In line 218, authors added to show their trials for an addition of different PEG and their concentration for FLAL. Just to clarify, wild-type protein DO NOT form any nanorod structure with addition of PEG you used for FLAL?

Minor point:

5. Regarding line 150, You better add information that 50 mM Ca^{2+} is necessary to form dimeric TmFTn into nanocages somewhere as you answered to my question (6). Reader would come up with same question as I did.

6. Regarding my question (11), please put yellow back if you think it helps to understand structure well. I asked because I didn't think you mention anything about yellow line.

7. You should mention protein size (M.W.) at least once in somewhere (ex. Captions of supp Fig 2). You mentioned it dimer formation in text, but it's not clear since no protein size mentioned anywhere. (same comment as of minor comment 9 previously)